# Association between *AHR* Expression and Immune Dysregulation in Pancreatic Ductal Adenocarcinoma: Insights from Comprehensive Immune Profiling of Peripheral Blood Mononuclear Cells

**DOI:** 10.3390/cancers15184639

**Published:** 2023-09-19

**Authors:** Arenida Bartkeviciene, Aldona Jasukaitiene, Inga Zievyte, Darius Stukas, Sandra Ivanauskiene, Daiva Urboniene, Toivo Maimets, Kristaps Jaudzems, Astra Vitkauskiene, Jason Matthews, Zilvinas Dambrauskas, Antanas Gulbinas

**Affiliations:** 1Laboratory of Surgical Gastroenterology, Institute for Digestive Research, Lithuanian University of Health Sciences, Eiveniu 4, 50103 Kaunas, Lithuania; aldona.jasukaitiene@lsmuni.lt (A.J.); inga.zievyte@lsmuni.lt (I.Z.); darius.stukas@lsmuni.lt (D.S.); sandra.ivanauskiene@lsmuni.lt (S.I.); zilvinas.dambrauskas@lsmuni.lt (Z.D.); antanas.gulbinas@lsmuni.lt (A.G.); 2Department of Laboratory Medicine, Lithuanian University of Health Sciences, Eiveniu 4, 50103 Kaunas, Lithuania; daiva.urboniene@lsmuni.lt (D.U.); astra.vitkauskiene@lsmuni.lt (A.V.); 3Department of Cell Biology, Institute of Molecular and Cell Biology, University of Tartu, Riia 23, 51010 Tartu, Estonia; toivo.maimets@ut.ee; 4Department of Physical Organic Chemistry, Latvian Institute of Organic Synthesis, Aizkraukles 21, LV-1006 Riga, Latvia; kristaps.jaudzems@osi.lv; 5Department of Nutrition, Institute of Basic Medical Sciences, Faculty of Medicine, University of Oslo, 1046 Blindern, 0317 Oslo, Norway; jason.matthews@medisin.uio.no; 6Department of Pharmacology and Toxicology, University of Toronto, Toronto, ON M5S 1A8, Canada

**Keywords:** AHR, PBMC, PDAC, PD1, PDL1, personalised, immunotherapy

## Abstract

**Simple Summary:**

This study investigated the role of aryl hydrocarbon receptor (*AHR*) expression in pancreatic ductal adenocarcinoma patients’ peripheral blood immune cells. Peripheral blood mononuclear cells (PBMCs) were collected from 30 pancreatic ductal adenocarcinoma patients and 30 healthy controls. Patients were divided into Low and High/Medium *AHR* groups based on *AHR* gene expression in PBMCs. The Low *AHR* group exhibited distinct immune features, including increased free PD1 and PDL1 protein levels, lymphocyte/monocyte subtype alterations, decreased phagocytosis, increased nitric oxide production and expressed cytokine imbalances, particularly IL-4. These findings showed a potential link between the expression of *AHR*s and immune dysregulation in patients with pancreatic ductal adenocarcinoma. AHR may play a role in modulating the immune response against cancer. The insights gained from these investigations have the potential to pave the way for the development of innovative approaches for the treatment of pancreatic ductal adenocarcinoma.

**Abstract:**

Pancreatic cancer, particularly pancreatic ductal adenocarcinoma (PDAC), has an immune suppressive environment that allows tumour cells to evade the immune system. The aryl-hydrocarbon receptor (AHR) is a transcription factor that can be activated by certain exo/endo ligands, including kynurenine (KYN) and other tryptophan metabolites. Once activated, AHR regulates the expression of various genes involved in immune responses and inflammation. Previous studies have shown that AHR activation in PDAC can have both pro-tumorigenic and anti-tumorigenic effects, depending on the context. It can promote tumour growth and immune evasion by suppressing anti-tumour immune responses or induce anti-tumour effects by enhancing immune cell function. In this study involving 30 PDAC patients and 30 healthy individuals, peripheral blood samples were analysed. PDAC patients were categorized into Low (12 patients) and High/Medium (18 patients) *AHR* groups based on gene expression in peripheral blood mononuclear cells (PBMCs). The Low *AHR* group showed distinct immune characteristics, including increased levels of immune-suppressive proteins such as PDL1, as well as alterations in lymphocyte and monocyte subtypes. Functional assays demonstrated changes in phagocytosis, nitric oxide production, and the expression of cytokines *IL-1*, *IL-6*, and *IL-10*. These findings indicate that *AHR*’s expression level has a crucial role in immune dysregulation in PDAC and could be a potential target for early diagnostics and personalised therapeutics.

## 1. Introduction

Pancreatic ductal adenocarcinoma (PDAC) is the third leading cause of cancer in both men and women [1]. The average age of incidences and mortality rates for pancreatic cancers vary across different regions of the world. The highest age-standardized incidence rates are in Europe and North America, and the lowest rates are in Africa and Central Asia [2]. Regional differences in the number of patients with pancreatic ductal adenocarcinoma appear to reflect differences in medical care, as well as racial genetic differences [3]. Still, the major factors contributing to poor outcomes are late diagnosis and unsuccessful treatment strategies [4]. Chemotherapeutic strategies become ineffective over time, and radical surgical intervention is only possible in up to 20% of diagnosed cases [5]. Although immunotherapy has shown promise in treating other types of cancer, there have been no significant breakthroughs in using it to treat PDAC [6]. This may be due to immunological barriers, such as insufficient immune activation and excess immune suppression [7]. The presence of tumour cells, immune cell subsets, and inflammatory cytokines can affect the immune balance in the peripheral blood of PDAC patients. To enhance the clinical effectiveness of existing immunotherapy approaches, a deeper comprehension of the biological mechanisms that drive immune evasion in cancer is necessary [8]. Cytokines play an important role as effector molecules in signalling and initiating immunological responses against cancer cells [9,10,11]. However, PDAC can evade immunological recognition through a variety of mechanisms. First, cancer cells can decrease immune recognition by downregulating antigen presentation pathways [12,13]. Recent studies have focused on targeting immune checkpoints, including the PD1/PDL1 pathway. Additionally, programmed cell death protein 1 (*PD1*) and its ligand programmed cell death protein ligand 1 (*PDL1*) are overexpressed on activated T cells and tumour-associated macrophages, respectively, resulting in the induction of immune tolerance [14,15]. Some studies suggest that substances from tumour cells shift the transcriptional programme of tumour-associated macrophages (TAMs) from an M1-like phenotype to an M2-like phenotype [16]. M1 macrophages have the potential to kill tumour cells and mediate antibody-dependent cellular cytotoxicity and phagocytosis. However, in many tumours, there is an increased level of M2 macrophages, which contribute to cancer progression and metastasis by promoting cancer cell survival and proliferation, angiogenesis, and suppressing innate and adaptive immune responses [17]. Despite the remarkable success of PD1/PDL1 inhibitors in treating certain cancers like melanoma and non-small-cell lung cancer [18,19,20], a significant proportion of patients fail to achieve long-term and sustained efficacy from this therapy [21,22,23]. Previous research indicates that only 10–30% of patients demonstrate a positive response, while the majority either exhibit no response or develop resistance to PD1/PDL1 inhibitors. Additionally, response rates in PDAC are rare [24]. Understanding the mechanisms underlying PD1/PDL1 antibody resistance is crucial for improving treatment outcomes and developing effective strategies to overcome this resistance. Due to reduced T cell infiltration and suppressed immunity in pancreatic cancer, the level of inflammation signal required to activate *PDL1* expression is also reduced. This means that PD1/PDL1 immunotherapy can be effective only in 50% of tumour cells, while the rest must be destroyed using other cancer treatment strategies. The indoleamine 2,3-dioxygenase-kynurenine (IDO-KYN) pathway has emerged as a significant focus of research in understanding cancer-immune escape mechanisms [25]. While numerous studies have investigated the regulatory effects of the IDO-KYN pathway, the importance of the aryl hydrocarbon receptor (AHR) as a cancer regulator has garnered increasing attention. It is known that surface *PD1* expression in T cells is regulated by AHR signalling [26]. In addition, numerous studies have demonstrated increased expression of *AHR* in various cancer subtypes, including chronic lymphocytic leukaemia [27], melanoma [28], breast cancer [29] brain cancer [30] and pancreatic cancer [31,32]. However, previous studies regarding pancreatic cancer have been conducted on cancer cells or immune cells in the tumour microenvironment (TME) [33]. To the best of our knowledge, the role and expression of *AHR* have not been studied in peripheral blood mononuclear cells (PBMCs) of cancer patients. However, they have been examined in some other inflammatory diseases: atopic dermatitis and morbid obesity [34,35]. Previous studies have not yet explored the changes in anticancer immunity and *AHR* expression, specifically in the blood of pancreatic cancer patients. PBMCs are readily available and can provide valuable insights into these aspects. Analysing PBMCs from PDAC patients can help shed light on alterations in *AHR* expression and its impact on anticancer immune responses. We hypothesise that the expression of *AHR* in PDAC patients’ PBMCs could affect and/or reflect anticancer immunity functions. Thus, we compared the function and the phenotypic composition of the PBMCs obtained from the peripheral blood of pancreatic cancer patients and healthy subjects.

## 2. Materials and Methods

### 2.1. Patient Population

PBMCs were isolated from the venous blood of patients with PDAC and healthy controls (Ctrl). The PDAC patient group had a mean age of 61.7 years (range: 37–87), consisting of 11 women and 19 men (Appendix A). The control group had a mean age of 50.8 years (range: 23–78), comprising 23 women and 7 men. The majority of PDAC cases were classified as Stage III and IV. Blood samples from 30 healthy donors with no previous cancer history were collected for the control group. In contrast, 30 samples were obtained from PDAC patients whose pancreatic cancer was histologically confirmed either during surgery or through biopsy.

The main inclusion criteria for this study group of cancer patients are:Clinically, radiologically or cytologically/histologically diagnosed pancreatic cancer;Planned surgical treatment of cancer.The patient is treated at LUHS Kaunas clinics.Age range of 20–90 years.

The main inclusion criteria for the control group are:No cancer diagnosed clinically, radiologically or cytologically/histologically.Not currently suffering from a viral or bacterial infection, have no signs of an inflammatory disease (not have a fever, not feel pain, fatigue or weakness).Not taking medications to affect the immune system: anti-inflammatory drugs, pain relievers or antibiotics.Age range of 20–90 years.

All samples were collected with informed consent from the patients, and the study protocol was approved by the Kaunas Regional Biomedical Research Ethics Committee (Approval Nr. BE-2-62).

### 2.2. PBMC Isolation and Cultivation

Peripheral blood collected into vacutainers with EDTA K2 (BD, Plymouth, UK) was immediately centrifuged at 2470× *g* for 10 min for plasma separation. PBMCs were isolated by Ficoll-Paque PREMIUM (Cytiva, Uppsala, Sweden) gradient centrifugation according to the manufacturer’s protocol [36]. The PBMCs from patients and healthy individuals were subjected to repeated centrifugation and dilution. The concentration of cells for functional activity analysis was maintained at 1 × 10^6^ cells/mL in Roswell Park Memorial Institute 1640 medium (RPMI) (Gibco Life Technologies Limited, Paisley, UK) without phenol red, along with 10% Fetal Bovine Serum (FBS) (Gibco Life Technologies Limited, Paisley, UK) and 1% penicillin/streptomycin (Gibco Life Technologies Limited, Paisley, UK). Subsequently, the samples were incubated at 37 °C for 30 min and distributed into dark or clear 96-well plates. Each well contained 90,000 cells. After a 3 h activation of PBMCs with various effectors, fluorometric or spectrophotometric assays were employed to measure phagocytosis, reactive oxygen species (ROS) or nitric oxide (NO) production.

### 2.3. Phagocytosis

Lipopolysaccharide (LPS) is a potent activator of PBMCs. It is a component of the outer membrane of Gram-negative bacteria and is recognized by Toll-like receptor 4 (TLR4) on immune cells. LPS stimulates the production of pro-inflammatory cytokines such as tumour necrosis factor alpha (TNF-α), IL-1β, and IL-6 in PBMCs. After LPS stimulation, macrophages and dendritic cells exhibit increased phagocytic activity, allowing them to efficiently engulf and eliminate pathogens or foreign substances. This enhanced phagocytosis is mediated by changes in cell surface receptors, cytoskeletal rearrangements, and increased production of phagocytic-related molecules [37]. To activate PBMCs before phagocytosis, 10 µL of 5 µg/mL LPS (Sigma-Aldrich, Rehovot, Israel) was added to three wells of a dark 96-well plate. Another three wells were used as negative controls with 10 µL of LPS diluent, and the last three wells served as medium controls with only RPMI and 10 µL of 5 µg/mL LPS without cells. Following the 2 h activation of PBMCs function, the dark plate was centrifuged, and the supernatant was removed. Next, 100 µL of pHrodo Green Zymosan Bioparticles (Invitrogen, Eugene, OR, USA) [38] was added to the dark plate for phagocytosis activation. The plate with bioparticles was incubated at 37 °C for an additional hour, followed by fluorescence measurements using a fluorimeter (Fluoroskan Ascent, Thermo Fisher Scientific, Waltham, MA, USA) with excitation at 510 nm and detection at 538 nm.

### 2.4. NO Production

L-arginine, an amino acid, modulates the effects of LPS on PBMCs. L-arginine can enhance the production of nitric oxide (NO) in response to LPS stimulation, which is involved in various immune functions. Additionally, L-arginine can regulate the balance between pro-inflammatory and anti-inflammatory cytokines, potentially influencing the immune response triggered by LPS [39]. NO production was measured after 3 h of treatment with 10 µL of 5 µg/mL LPS + 100 Mm L-arginine (Sigma-Aldrich, Tokyo, Japan) in a clear 96-well plate. LPS + L-arginine was added to three wells, while another three wells served as negative controls with 10 µL of LPS + L-arginine diluent. Additionally, in three wells, only RPMI with 10 µL of 5 µg/mL LPS + 100 Mm L-arginine without cells was added as a medium control. Following 2.5 h activation at 37 °C, 100 µL of Griess reagent (Invitrogen, Eugene, OR, USA) [40] was added to the clear plate. Next, the plate with Griess reagent and samples were incubated at 37 °C for half an hour, followed by spectrophotometric measurements using a spectrophotometer (The Sunrise, Tecan, Grodig, Austria) at an absorption of 550 nm.

### 2.5. ROS Production

Tert-butyl hydroperoxide (TBHP) is an organic peroxide and oxidative-stress-inducing agent. It can cause cellular damage by generating reactive oxygen species (ROS) in PBMCs. TBHP is used as a model compound to study oxidative-stress-related cellular responses. Its effects on PBMCs include oxidative damage, activation of stress response pathways, and modulation of immune cell functions. ROS production was measured after DCFDA (Cellular ROS Assay Kit, Abcam, Cambridge, UK)-stained PBMCs were activated after 3 h of treatment with 10 µL of 2.5 mM TBHP (Abcam, Cambridge, UK) in a dark 96-well plate according to the manufacturer’s protocol [41]. TBHP was added to three wells, while another three wells served as negative controls with 10 µL of TBHP diluent. Additionally, in three wells, only RPMI with 10 µL of 2.5 mM TBHP without cells was added as a medium control. The plate was then incubated for 3 h at 37 °C followed by fluorescence measurements using a fluorimeter (Fluoroskan Ascent) with excitation at 485 nm and detection at 538 nm.

### 2.6. RNA Extraction and Real-Time Polymerase Chain Reaction (RT-PCR)

Total RNA extraction from PBMCs was performed using the RNA extraction kit (Abbexa, Cambridge, UK) following the manufacturer’s protocol. The purified RNA was quantified and assessed for purity using spectrophotometry (NanoDrop 2000, Thermo Fisher Scientific, Waltham, MA, USA). Subsequently, cDNA was synthesised from 2 μg of RNA using the High/Medium-Capacity cDNA Reverse Transcription Kit (Applied Biosystems, Waltham, MA, USA). For the amplification, a 20 μL reaction mixture was prepared, which included 2 μg of cDNA template, 1 × PCR master mix, and primers. The primers used for PCR detection of *AHR* (Hs00169233_m1), *Interleukin (IL)-1b* (Hs001555410_m1), *IL-4* (Hs00174122_m1), *IL-6* (Hs00174131_m1), *IL-10* (Hs00961619_m1), *PD1* (Hs05043241), *PDL1* (Hs00204257), and the housekeeping gene *GAPDH* (Hs02786624) were obtained from Applied Biosystems (Waltham, MA, USA).

### 2.7. Western Blot Analysis

Cells were lysed using radioimmunoprecipitation assay buffer (RIPA) containing protease inhibitors (Roche, Basel, Switzerland). Lysates were then centrifuged at 10,000× *g* for 10 min. Supernatants were collected and used for protein concentration determination using the BCA protein assay kit (Thermo Fisher Scientific, Waltham, MA, USA). Next, 45 µg of protein samples with Bolt MES SDS loading buffer (Invitrogen, Carlsbad, CA, USA) was heated at 97 °C for 5 min and loaded onto a 4–12% sodium dodecyl sulphate-polyacrylamide gel electrophoresis (SDS-PAGE) and transferred onto poly-vinylidene fluoride (PVDF) membranes for 40 min at 30 V. Then, the membranes were blocked with a 5% blocking buffer (Invitrogen, Waltham, MA, USA) for 40 min at room temperature. Next, the membranes were incubated with primary antibodies for 1 h at room temperature or overnight at 4 °C. The primary antibodies used included a 1:1000 dilution of mouse monoclonal anti-AHR (MA1-514, Invitrogen, Waltham, MA, USA), a 1:1000 dilution of rabbit monoclonal anti-PD1 (SJ01-91, Invitrogen, Waltham, MA, USA), and a 1:3000 dilution of mouse monoclonal anti-GAPDH (AM4300, Invitrogen, Waltham, MA, USA). The membranes were washed and incubated with the appropriate horseradish-peroxidase-conjugated secondary antibody (HRP) (Invitrogen, Carlsbad, CA, USA) for 1 h or 30 min at 37 °C. Following another round of washing, the membranes were incubated with a chemiluminescence substrate (Invitrogen, Carlsbad, CA, USA) or West Pico Stable peroxidase buffer with luminol enhancer (Thermo Scientific, Waltham, MA, USA) for 5 min. The results were analysed using a ChemiDoc Imaging System (Bio-Rad Laboratories, Hercules, CA, USA) and ImageJ software 1.53a (National Institutes of Health, Bethesda, MD, USA).

### 2.8. Flow Cytometry (Lymphocyte Subsets, Monocyte/Macrophage Subsets)

Flow cytometric immunophenotyping using the BD multi-test 6-colour T and B Natural Killer Cells (TBNK) kit (BD, Franklin Lakes, NJ, USA) was performed to determine the following lymphocyte subsets: CD19+ (B lymphocytes), CD3+ (T lymphocytes), CD3+/CD4+ (T helpers), CD3+/CD8+ (T cytotoxic lymphocytes), and CD3−/CD16+ 56 + (natural killer (NK) cells). For this purpose, aliquots of PBMC (1 million cells each) were incubated for 10 min at room temperature with combinations of the following monoclonal antibodies (mAbs): CD3-FITC (Leu-4), CD4 PE-Cy™7 (Leu-3a), CD8 APC-Cy™7 (Leu-2a), CD16 PE (Leu-1 lc), CD19 APC (Leu-12), CD56 PE (Leu-19), and CD45 PerCP-Cy5.5 (2D1) (BD, USA). After incubation, the cells were washed and subsequently analysed using a 10-colour flow cytometer FACSLyric (BD, San Jose, CA, USA). Up to 10,000 total events were recorded per sample. The forward/side scatter and fluorescence signals provided information about the cell size, internal complexity, and relative fluorescence intensity. The lymphocyte population was selected based on granularity/complexity (side scatter—SSC) and the CD45 expression level. The number of cells with positive expression of tested antigens was reported as a percentage of the cells in the lymphocyte gate. Phenotypic analyses of PBMCs for discrimination of monocyte/macrophage subsets were conducted using the following mAbs: CD3 PE (UCHT1), CD14 BV510 (MφP9), CD16 APC (B73.1), CD80 APC-H7 (L307.4), CD86 PE-Cy7 (2331 (FUN-1)), CD163 BV605 (GHI/61), CD206 FITC (19.2), HLA-DR PerCP (L243), CD19 PE (HIB19), CD56 PE (555516), and CD66b PE (G10F5). These antibodies allowed for the identification and quantification of specific cell populations. To determine the cell populations responsible for cytokine production, the following antibodies were used: IL-1b (H1b-98) Pacific Blue (BioLegend, USA), IL-4 (MP4-25D2), and IL-6 (MQ2-13A5). They were detected using BV421, BV421, and IL-10 (JES3-19F1) (BD, USA). The promotion of intracytoplasmic cytokine accumulation in the cells was conducted using BD Cytofix/Cytoperm™ Plus Fixation/Permeabilization Kit (with BD GolgiStop™ protein transport inhibitor containing monensin) (BD, USA) by adding 1 µL of BD GolgiStop™ (Cat. No. 554715, BD, USA) to 1 million PBMC for the 4 h treatment. Multicolour staining for cell surface antigens and intracellular cytokines was conducted according to the BD protocol provided with the BD Cytofix/Cytoperm™ Plus Fixation/Permeabilization Kit (USA). PMT voltage and compensation were set using BD™ CompBeads Anti-Mouse Ig, κ/Negative Control Compensation Particles Set (Cat. No. 552843, BD, USA). Up to 30,000 total events were collected per sample for monocyte/macrophage subsets and their cytokine production analysis. Cells of interest were selected on the SSC/CD45 plot. The cells of no interest (lymphocytes and residual granulocytes) were excluded using their typical immunophenotype characteristics. The M1 subset was identified by the combination of markers CD45+/CD80+/CD86+ and M2 by CD45+/CD163+/CD206+. Monocyte/macrophage subsets are reported as a percentage of the cells in the lymphocyte gate, and their cytokine expression is presented using mean fluorescence intensity (MFI) values converted to percentages.

### 2.9. Luminex (Concentrations of Serum Cytokines)

The concentrations of serum cytokines—IL-1b, IL-4, IL-6 and IL-10—were quantified using magnetic bead-based multiplex assays (Human Cytokine Premixed Multi-Analyte Kit, R&D) by a Luminex^®^ 100 analyser (Luminex Corporation, Austin, TX, USA). Peripheral blood samples were collected into 3 mL vacutainers with SST (serum clot activator) (BD, Plymouth, UK) and centrifuged at 2500× *g* at 4 °C for 20 min. Afterwards, the serums were carefully collected, transferred into cryogenic tubes and stored at −80 °C until use. On the day of testing, previously frozen serum samples were thawed and centrifuged at 16,000× *g* at 4 °C for 4 min to remove any debris or precipitates. The samples were processed according to the manufacturer’s protocol. Analyte-specific antibodies were pre-coated onto magnetic microparticles embedded with fluorophores at set ratios for each unique microparticle region. Microparticles, standards and samples were pipetted into wells, and the immobilised antibodies were bound to the target substances. After washing away any unbound substances, samples were incubated with a mixture of biotinylated detection antibodies and a streptavidin–phycoerythrin (SAPE) reporter. Final washes were used to remove unbound SAPE, and the microparticles were resuspended in buffer and analysed using the Luminex^®^ 100 analyser. Using a Luminex instrument, beads were excited by the laser to determine the bead region and corresponding assigned analyte. The magnitude of the PE-derived signal, which was proportional to the amount of analyte bound, was obtained by another laser. Multiple MFI readings were taken at each bead region, ensuring robust detection. The values of cytokine concentrations were assigned in relation to the standard curve (plotting the MFI against the protein concentration).

### 2.10. ELISA (Concentrations of PD1, PDL1 and KYN)

Blood samples were centrifuged at 2500× *g* at 4 °C for 20 min. The resulting sera were carefully collected and transferred into cryogenic tubes, which were then stored at −80 °C until the day of the enzyme-linked immunosorbent assay (ELISA). The concentration of PD1 (Human PD1 ELISA Kit ab252360, Abcam, Cambridge, MA, USA), PDL1 (Human PDL1 ELISA Kit ab277712, Abcam, Cambridge, MA, USA) and KYN (EKC34358, Biomatic, Boston, MA, USA) in the supernatants was determined using a quantitative ELISA. The assay was carried out according to the manufacturer’s protocols. Calibration curves were plotted, and concentrations of PD1, PDL1 and KYN in blood samples were determined by interpolation from the calibration curves by comparing the optical density of the samples.

### 2.11. Statistical Analysis

Statistical analysis was performed using GraphPad Prism software (version 9.01; GraphPad Software Inc., La Jolla, CA, USA). The data were presented as medians and visualised in scatter plots, displaying the medians with interquartile ranges. Western blot data were presented as means with standard deviation (+/− SD) and visualised in bar plots. For the comparison between groups, nonparametric Mann–Whitney (two-tailed) and Kruskal–Wallis tests with Dunn’s multiple comparisons test were employed. Statistical significance was defined as *p* < 0.05. To assess the strength of the association, the Spearman rank correlation was utilised as a measure of monotonic association. The mRNA expression values of PDAC patients were divided into two groups based on the 33rd and 67th percentiles (0–33rd and close to 33rd percentiles representing the first group and above the 34–100th percentile representing the second group).

## 3. Results

### 3.1. AHR Level in PBMCs of PDAC Patients

The trend of mRNA expression of *AHR* was lower in the PBMCs of PDAC patients compared to healthy controls (Figure 1; PDAC = 0.76 vs. Ctrl = 1.18), even though it was without statistical significance. Interestingly, within the PDAC patient population, a specific subgroup was identified (represented by blue dots) exhibiting notably lower levels of *AHR* mRNA expression compared to others. This subgroup was named the Low *AHR* group. Conversely, the remaining PDAC patients (represented by red dots) demonstrated relatively higher levels of *AHR* mRNA expression and were categorised as the High/Medium *AHR* group.

The Low *AHR* group of patients was exceptionally clinical. All 12 Low *AHR* group patients had more advanced tumours (T3 and T4 stage), whereas 8 out of 18 patients from the High/Medium *AHR* group had early stage (T1 and T2) tumours (Figure 2a). In total, 8 out of 12 tumours in the Low *AHR* group were well differentiated (G1 or G2). A similar proportion was observed in the High/Medium *AHR* group; 11 out of 18 were G1 or G2 tumours (Figure 2b).

The only effective treatment possible for pancreatic cancer is radical surgery. There was a very important finding in our cohort of patients: 14 of 18 High/Medium *AHR* patients underwent a radical operation, whereas in 11 out of 12 of the Low *AHR* group, surgery was not possible (Figure 3a). That was because most Low *AHR* patients had either locally advanced or systematically spread disease at the time of diagnosis (Figure 3b).

Interestingly, pancreatic cancer patients were statistically significantly younger in the Low *AHR* group (Figure 4a), although the gender proportion remained similar regardless of the *AHR* level (Figure 4b).

Based on the normalised data obtained from Western blot analysis, it was observed that the protein level trend of AHR was also found to be lower in the Low *AHR* group of patients (Low = 16%, SD = 11 vs. High/Medium = 776%, SD = 1227; Ctrl = 100%, SD = 96) (Figure 5a,b), although it was not statistically significant.

### 3.2. PD1 and PDL1 Levels in PDAC Patients’ Blood Depending on AHR Level

In some cancers, it has been shown that AHR activation can contribute to the upregulation of *PD1* and *PDL1* expression in immune cells, such as T lymphocytes and monocytes, leading to immune suppression and tumour immune evasion. AHR activation can also affect the function and activity of regulatory T cells (Tregs), influencing the expression of immune checkpoint molecules like PD1 [42]. AHR activation can also affect the function and activity of regulatory T cells (Tregs), influencing the expression of immune checkpoint molecules like PD1. Therefore, we investigated the possible link between the expression of *PD1/PDL1* and *AHR* genes in PBMCs of pancreatic cancer patients. The results demonstrated that the trend of *PD1* mRNA expression was increased in PBMCs of PDAC patients compared to healthy controls (Figure 6a) (PDAC = 1.2 vs. Ctrl = 1.08, *p* = 0.75). However, the difference was not statistically significant. Conversely, *PDL1* mRNA expression was significantly decreased in PDAC patients (PDAC = 0.6, Ctrl > 0.999, *p* = 0.01) compared to healthy controls. Furthermore, the Spearman correlation analysis revealed associations between *PD1*, *PDL1* and *AHR* gene expression in PBMCs of all PDAC patients (Figure 6b). Specifically, *AHR* displayed a strong positive correlation with *PDL1* (r = 0.84, *p* < 0.01), indicating a highly significant relationship. On the other hand, PD1 exhibited a weak association with *AHR* (r = 0.35, *p* > 0.07).

After patient grouping by *AHR* level, the expression trend of *PD1* was decreased in the PBMCs of the Low *AHR* group, whereas in the High/Medium *AHR* group, it was increased (Figure 7a) (Low = 0.65; High/Medium = 1.51; Ctrl = 1.08; *p* > 0.05). However, this difference did not reach statistical significance. Despite that, the mRNA expression of the *PDL1* gene was significantly decreased in Low *AHR* group (Figure 7b) (Low = 0.29; High/Medium = 0.81; Ctrl > 0.999) compared to both the healthy control group (*p* < 0.01) and the High/Medium *AHR* group (*p* < 0.01). These results indicate a significant downregulation of *PDL1* expression in the Low *AHR* group compared to both the healthy control and the High/Medium *AHR* group.

After these results, we analysed PD1 and PDL1 protein levels in the rest of the samples. The normalised data from the protein analysis showed that the trend of PD1 protein levels was also lower in the Low *AHR* group of patients (Low = 24%, SD = 7 vs. High/Medium = 1082%, SD = 1678; Ctrl = 100%, SD = 128) (Figure 8a,b) without statistical significance.

Conversely, the level of free PD1 protein was significantly increased in the Low *AHR* group (Figure 9a) (152 pg/mL) compared to the High/Medium *AHR* group (51.5 pg/mL; *p* < 0.01) and showed no significant difference compared to the healthy control (128 pg/mL, *p* = 0.16). Additionally, the level of PDL1 was significantly increased in the Low *AHR* group (Figure 9b) (121.5 pg/mL) compared to the High/Medium *AHR* group (57 pg/mL, *p* = 0.031) and the healthy control (78.5 pg/mL, *p* = 0.047).

### 3.3. PBMC Population Composition in PDAC Patients’ Blood by AHR Level

The results of PD1 and PDL1 in PBMC cells and blood serum raise questions about the differences in the subpopulation of PBMC cells depending on the *AHR* level. The results from the classification of the total population of PBMCs (Figure 10a) demonstrated a significant decrease in the percentage of T cytotoxic cells in the Low *AHR* group compared to the healthy control group (Low = 17.95% vs. High/Medium = 20.68%, *p* = 0.5; Ctrl = 31.38%, *p* < 0.01). Additionally, a significant decrease in the percentage of M0 type monocytes (Figure 10b) was observed in the Low *AHR* group compared to both the healthy control group (Low = 0.95% vs. High/Medium = 1.74%, *p* = 0.1; Ctrl = 1.68%, *p* < 0.01). Conversely, there was a significant increase in the population of B cells (Low = 9.84% vs. High/Medium = 7.81%, *p* = 0.1; Ctrl = 3.68%, *p* < 0.01) and NK cells (Low = 18.93% vs. High/Medium = 16.28%, *p* = 0.3; Ctrl = 11.79%, *p* < 0.01) in the Low *AHR* group compared to the healthy control group.

### 3.4. PBMC Functional Activity Dependence on AHR Level in PDAC Patients

In the presence of differences in the composition of the PBMC populations, unevenness in the activity of functions should also appear. The functional activity of phagocytosis and ROS production in PBMCs was assessed and compared between healthy controls and different *AHR* expression groups of PDAC patients. Phagocytosis (Figure 11a) was observed to decrease in all PDAC patients, with a significant decrease specifically in the Low *AHR* group compared to the healthy control group (Low = 64%; High/Medium = 76%, *p* = 0.057; Ctrl = 96%, *p* < 0.01). Additionally, NO production (Figure 11B) was increased in the Low *AHR* group, while the High/Medium *AHR* group trend was decreased compared to the healthy control group (Low = 115%; High/Medium = 76%, *p* = 0.038; Ctrl = 94%, *p* = 0.037). However, the trend in ROS production (Figure 11c) was decreased in the Low *AHR* group, while the High/Medium *AHR* group showed a slight increase compared to the healthy control group (Low = 95%; High/Medium = 102%, *p* = 0.8; Ctrl = 98%, *p* = 0.6).

### 3.5. KYN Level in PDAC Patients’ Serum Depending on AHR Level

The trend of KYN was decreased in the Low *AHR* group (Figure 12) (238 nmol/mL) compared to the High/Medium *AHR* group (250 nmol/mL, *p* = 0.09). However, the level of KYN was significantly increased in the Low AHR group compared to the healthy control (224.5 nmol/mL, *p* = 0.019).

### 3.6. IL-1b, IL-4, IL-6 and IL-10 Differences in PDAC Patients’ Blood by AHR Level

Although monocytes were more differentiated in the Low *AHR* group, the phagocytosis function was impaired, and the cytokine background might be responsible for this. The median fold change in *IL-1b*, *IL-4*, and *IL-6* mRNA expression (Figure 13a) was found to be significantly downregulated in PDAC PBMCs compared to healthy controls (*IL-1b*: PDAC = 0.2 vs. Ctrl = 1.33, *p* < 0.01; *IL-4*: PDAC = 0.23 vs. Ctrl = 1.015, *p* < 0.01; *IL-6*: PDAC = 0.29 vs. Ctrl = 1.42, *p* = 0.01). On the other hand, *IL-10* mRNA expression was upregulated in PDAC, but the change was not statistically significant compared to healthy controls (*IL-10*: PDAC = 1.23 vs. Ctrl = 0.84, *p* = 0.958). Analysis among PDAC patients also revealed relationships between *AHR* expression and the measured cytokine mRNA expressions (Figure 13b). *AHR* showed a very strong positive influence on *IL-1b* (r = 0.81, *p* < 0.01), IL-6 (r = 0.87, *p* < 0.01), and *IL-10* (r = 0.85, *p* < 0.01), and a moderate positive influence on IL-4 (r = 0.40, *p* = 0.031). The correlations between IL-4 and other target gene mRNAs were weak: *IL-1b* (r = 0.22), *IL-6* (r = 0.30), and *IL-10* (r = 0.28). Strong positive associations were observed among all other cytokines in the PDAC mRNA samples.

The Low *AHR* group exhibited remarkably low expression levels of all measured cytokine mRNAs compared to the other PDAC group and healthy controls. *IL-1b* mRNA expression (Figure 14a) was significantly dysregulated in the Low *AHR* group (Low = 0.04 vs. High/Medium = 0.48, *p* < 0.01; Ctrl = 1.33, *p* < 0.01). Additionally, *IL-6* cytokine mRNA (Figure 14c) expression was significantly decreased in the Low *AHR* group (Low = 0.08 vs. High/Medium = 0.65, *p* < 0.01; Ctrl = 1.42, *p* < 0.01). Similarly, *IL-10* mRNA expression (Figure 14d) was significantly downregulated in the Low *AHR* group (Low = 0.32 vs. High/Medium = 2.04, *p* < 0.01 Ctrl = 0.84, *p* = 0.012). Only *IL-4* mRNA expression in the Low *AHR* group was not significantly different from the other groups (Low = 0.18 vs. High/Medium = 0.24, *p* = 0.17; Ctrl = 1.015, *p* < 0.01) corresponding to the data presented in the correlation matrix (Figure 14b).

Although the expression of *IL-4* mRNA in the Low *AHR* group was similar to the other groups, the concentration of *IL-4* protein in the serum (Figure 15b) significantly increased in the same group (Low = 131 pg/mL vs. High/Medium = 103.5 pg/mL, *p* < 0.01; Ctrl = 108 pg/mL, *p* = 0.02). There were no differences observed in *IL-1b* (Figure 15a) and *IL-10* (Figure 15d) protein levels among all other groups (*IL-1b*: Low = 117 pg/mL vs. High/Medium = 117 pg/mL, *p* = 0.3; Ctrl = 120.5 pg/mL, *p* = 0.8; *IL-10*: Low = 50 pg/mL vs. High/Medium = 49 pg/mL, *p* = 0.4; Ctrl = 50 pg/mL, *p* = 0.7). However, *IL-6* (Figure 15c) concentration in the serum was increased in PDAC patients compared to healthy controls (Low = 42 pg/mL vs. High/Medium = 42 pg/mL, *p* = 0.7; Ctrl = 35.3 pg/mL, *p* < 0.01).

The results obtained from the analysis of protein secretion on monocytes indicated that the expression of IL-4 (Figure 16b) and IL-6 (Figure 16c) cytokines on the surface of monocytes was not dependent on *AHR* level (no significant difference between all groups). However, IL-1b secretion (Figure 16a) was significantly increased in both groups of PDAC patients on M1 type monocytes (M1: Low = 120% vs. High/Medium = 121%, *p* > 0.999; Ctrl = 101%, *p* = 0.04; M2: Low = 111% vs. High/Medium = 113%, *p* > 0.999; Ctrl = 101%, *p* > 0.999). Interestingly, IL-10 secretion (Figure 16d) was downregulated significantly in M2-type monocytes in the Low *AHR* group. However, in M1, it was downregulated in the High/Medium *AHR* group (M1: Low = 38% vs. High/Medium = 31%, *p* > 0.999; Ctrl = 98%, *p* = 0.051; M2: Low = 39% vs. High/Medium = 36%, *p* > 0.999; Ctrl = 99%, *p* = 0.04).

## 4. Discussion

### 4.1. Low Expression of AHR in PDAC Patients Is Associated with Altered Immune Responses and Imbalances in Immune Checkpoint Molecules

The immune system plays a vital role in maintaining the delicate balance between the immune response and tumour development. Our study found a trend of *AHR* mRNA downregulation in the PBMCs of patients with PDAC compared to healthy controls. Although *AHR* expression varied largely in healthy controls and PDAC patients’ PBMCs, there appeared to be a subgroup of patients with distinctively lower *AHR* expression. Previous studies have shown that AHR activation can have both pro-tumorigenic and anti-tumorigenic effects depending on the context [43]. The downregulation of AHR in PBMCs of PDAC patients may have implications for the immune response against the tumour. Identifying a particular subgroup of PDAC patients with lower *AHR* expression suggests that the AHR pathway may be dysregulated in these patients. The normalised data from the Western blot analysis indicates that AHR protein levels are lower in the Low *AHR* group of patients compared to controls and the High/Medium *AHR* group. This finding is particularly interesting in earlier studies, as AHR had been shown to play a crucial role in regulating cellular processes such as proliferation, differentiation, and apoptosis, all of which are perturbed in cancer cells [44]. One possible explanation for this observation is that lower *AHR* expression in PBMCs might have a different AHR background in cancer tissue. *AHR* expression can be modulated by various environmental factors or endogenous ligands. In earlier studies on human PDAC tissue samples, increased *AHR* expression was associated with rapid disease progression, higher mortality, and an immune-suppressive TAM phenotype [30,45]. Several studies have investigated the main mechanisms of AHR-regulated cancer development and the relationship between abnormal *AHR* expression and tumours [44,46,47,48]. Despite its role as an oncogene, AHR also acts as a tumour suppressor in several types of cancer, such as those affecting the brain and central nervous system, liver, digestive system, skin (melanoma), and reproductive system. Inhibitory roles of AHR have been revealed using engineered mouse models in which *AHR* expression was completely abrogated (*AHR*−/− mice). In these knock-out mouse models, liver tumorigenesis and growth were significantly higher than in control mice [49]; this may have been, at least in part, due to AHR missing in immune cells. Similar results have been observed in a model of colorectal tumorigenesis [50]. Changes in *AHR* expression in immune cells of the circulating systems of patients have shown a direct correlation between *AHR* and cytokines in atopic dermatitis and morbid obesity [34,35]. Our study focused on changes in the peripheral blood system of pancreatic cancer patients and identified a group of patients who, possibly, may not respond effectively to PD1/PDL1 inhibitors.

### 4.2. PDAC Patients with Lower AHR Expression Exhibit Decreased PD1 and Significantly Decreased PD-L1 mRNA Expression, Indicating a Potential Dysregulation of the PD1/PD-L1 Pathway in Tumour Immunity

Recent studies have suggested a potential connection between AHR and PD1/PDL1 regulation in the context of lung cancer [51]. However, previous studies have shown that aryl hydrocarbon signalling can increase T cell *PD1* expression [52]. Our study results confirmed a weakly positive trend between *AHR* and *PD1* mRNA expression in the peripheral blood of PDAC patients. Specifically, the data indicate that the trend of *PD1* expression decreased in the Low *AHR* group while it increased in the PBMCs of the High/Medium *AHR* group. However, this difference was not statistically significant. Additionally, we found that the expression of the *PDL1* mRNA was significantly decreased in the Low *AHR* group compared to the High/Medium *AHR* group and healthy controls. PD1 modulates T cell activity, activates antigen-specific T cell apoptosis, and inhibits regulatory T cell apoptosis, thereby inhibiting the immune response and promoting self-tolerance [53]. PDL1 can combine with PD1 to reduce the proliferation of PD1-positive cells, inhibit their cytokine secretion, and induce apoptosis [53,54]. Also, PDL1 is expressed in dendritic cells and reduces antigen presentation and T cell activation from naive lymphocytes [55]. Free soluble PDL1 in the serum of cancer patients has been linked to poor overall survival in several types of cancer [56,57,58]. Recent studies have shown that higher *PDL1* expression levels have been shown to predict poor prognosis in PDAC patients [59]. A high pre-therapeutic PDL1 expression in monocytes from blood has been detected as an adverse factor for PD1 inhibitor therapy in an earlier study of lung cancer [42]. In addition, there is a direct correlation between *PDL1* expression and infiltration of immune cells [60]. PD1/PDL1 immunotherapy blocks PDL1 in TAMs, but PDL1 is poorly expressed in PDAC tumour cells [61]. The effectiveness of anti-PD1/PDL1 antibodies in treating PDAC is thought to be due to a deficiency in PDL1 [62]. However, oncogenic or inflammatory cytokines have been shown to activate PDL1 expression in the tumour, especially interferon-gamma. Our study found a moderate correlation between *PD1* and *AHR* expression in PBMCs of PDAC patients, while there was a very strong relationship between *PDL1* and *AHR* expressions in patients’ PBMCs. Both *PD1* and *PDL1* expressions were decreased in the Low *AHR* group. However, PD1 and PDL1-free protein concentrations were increased in serum from the Low *AHR* group, while PD1 protein was highly decreased in PBMCs from High/Medium *AHR* patients.

### 4.3. Low AHR Group Demonstrates Decreased Expression of T Cytotoxic Lymphocytes and M0 Type Monocytes, Suggesting a Potential Impairment in the Effector Arm of the Immune Response. B Lymphocytes and NK Cells Are More Abundant in the Low AHR Group, Indicating Potential Compensatory Mechanisms in the Immune System

Many malignancies, including pancreatic cancer, can evade the immune system by indirectly regulating the defence mechanisms of PBMCs [6,10,12]. These cells, which include T and B lymphocytes, monocytes and NK cells, work as a network of complicated immunity chains. The flow cytometry data from our study revealed no significant changes in the population of monocytes and lymphocytes. However, when the cellular composition of the PBMC population was analysed, it was found that the percentage of cytotoxic T cells and M0 type monocytes were significantly decreased in the Low AHR group compared to the healthy control. On the other hand, there was a significant increase in the population of B and NK cells in this group compared to the healthy control. Previous studies have suggested that a higher T cytotoxic/T helper ratio may be associated with improved survival outcomes in PDAC patients [63]. The decrease in M0-type monocytes suggests impaired monocyte differentiation, which may contribute to the development and progression of PDAC. On the other hand, B and NK cells are key components of the anti-tumour immune response [64]. B cells produce antibodies that can recognise and neutralise cancer cells, while NK cells directly kill cancer cells [64,65]. The genotypic and phenotypic diversity of the cells present in a pancreatic tumour change throughout tumorigenesis, and PDAC is characterised by a dense mechanical fibroblast barrier [66,67]. This tumour instability and diverse regulatory mechanisms can lead to the loss of tumour antigens recognised by cytotoxic T and NK cells [12], resulting in the loss of effector cell subtypes and promoting tumour progression. The composition of PBMC populations can significantly impact phagocytosis and ROS-producing abilities.

### 4.4. Phagocytosis and ROS Production Are Decreased but NO Production Is Increased in the Low AHR Group, Reflecting Impaired Phagocytic and Oxidative Burst Activities of Immune Cells

In our study, phagocytosis was weak in all PDAC patients’ groups, but it was the weakest in the Low *AHR* group. Additionally, ROS production has the lowest trend in this group of PDAC patients. Previous studies have shown that phagocytosis is impaired in PDAC patients, and this impairment is associated with poor prognosis and survival rates [68]. The mechanisms behind this impairment are not fully understood, but it is thought to be related to the dysregulation of several signalling pathways and immune mediators, such as IL-10 [69]. Similarly, ROS production by monocytes is also impaired in PDAC patients, which can lead to decreased tumour cell killing and increased tumour growth [70]. NO, generated by inducible nitric oxide synthase (iNOS), is a multifunctional molecule involved in both physiological and pathological processes. In the context of cancer, NO has been implicated in tumour growth, angiogenesis, and immune evasion [71]. The increased NO production observed in the PBMCs of the Low *AHR* group suggests that NO may contribute to the immunosuppressive tumour microenvironment in PDAC. Elevated NO levels can suppress T cell responses, impair antigen presentation, and promote the accumulation of myeloid-derived suppressor cells (MDSCs) and regulatory T cells [72]. Furthermore, the enhanced NO production in the Low *AHR* group may be associated with the activation of alternative signalling pathways. For instance, IDO-TRP-KYN signalling, which might be indirectly modulated by antigen-presenting cells in NO, increased production [73,74]. However, in a recent study, we observed that NO can positively or negatively regulate IDO1 and TDO activities in cells through its influence on cell heme allocation [75]. The dysregulation of AHR-mediated pathways may disrupt the balance between pro-inflammatory and anti-inflammatory responses, favouring tumour progression and immune evasion but it is still not possible to determine which path is the most important.

### 4.5. Lack of Direct Connection between Kynurenine Level and AHR Expression in PDAC Patients’ Serum

Recent studies have reported a positive correlation between kynurenine levels and AHR expression, indicating that IDO-KYN pathway may activate AHR signalling in PDAC [76]. Kynurenine (KYN) is a metabolite produced through the breakdown of tryptophan, an essential amino acid. In the tumour microenvironment, increased levels of kynurenine have been observed in various cancer types, including PDAC, colorectal cancer, and melanoma [77]. Kynurenine exhibits immunosuppressive properties by suppressing T cell responses and promoting the expansion of regulatory T cells. These immunosuppressive effects of kynurenine contribute to tumour immune escape and facilitate tumour progression. Indoleamine 2,3-dioxygenase (IDO) is a key enzyme involved in the conversion of tryptophan to kynurenine. IDO expression is upregulated in various cancer types and is associated with poor prognosis [78]. Our findings reveal increased kynurenine levels in both Low and High/Medium AHR patient groups’ serum. Crosstalk between these pathways may influence the overall tumour phenotype and disease progression. The kynurenine pathway, IDO, and AHR pathway have gained significant attention in cancer research due to their potential roles in tumour progression and immune regulation. Tumour cells, as well as immune cells within the tumour microenvironment, can express IDO, leading to increased kynurenine production. IDO-mediated tryptophan depletion and kynurenine accumulation contribute to immune tolerance and suppression, impairing the anti-tumour immune response. In cancer patients’ blood, the interplay between kynurenine, IDO, and AHR pathway components can have significant implications. In recent studies, the dysregulated kynurenine metabolism, IDO expression, and AHR pathway activation in cancer patients’ blood contribute to immune evasion and tumour progression. However, IDO inhibitors do not increase the effectiveness of anti-PD-1/PD-L1 antibodies for T cell-inflamed tumours such as PDACs treated with GVAX therapy (GVAX^®^ is an investigational allogeneic cancer vaccine platform based on the local production of granulocyte-macrophage colony-stimulating factor (GM-CSF) at the vaccine site to generate a systemic tumour-specific immune response) [77].

### 4.6. Low AHR Group Shows Significantly Reduced Expression of Pro-Inflammatory Cytokines, Such as IL-1b, IL-4, and IL-6, at the mRNA Level, Suggesting an Immunosuppressive Microenvironment. IL-4 Protein Levels Are Increased in the Serum of the Low AHR Group despite Lower IL-4 mRNA Expression, Indicating Post-Transcriptional Regulation of IL-4 Production

This study conducted a correlation analysis of target gene expression data in PBMCs of PDAC patients, revealing a strong positive relationship between *AHR* and *IL-1b*, *IL-6* and *IL-10* expression. The findings showed that the median fold change in *IL-1b*, *IL-4* and *IL-6* expression was significantly downregulated in PDAC patients’ PBMC, particularly in the lower *AHR* group compared to other PDAC patients and healthy controls. Interestingly, IL-4 levels were significantly upregulated in serum from the Low *AHR* group, possibly regulated by monocytes or the TME. IL-4 has a dual role in regulating immunity and has been shown to promote tumorigenesis when acting synergistically with other cytokines [79]. Serum levels of IL-1b, IL-6 and IL-10 did not change significantly in the Low *AHR* group. IL-1b is a pro-inflammatory mediator often upregulated in various types of cancer, and its production is associated with a poor prognosis [80,81]. IL-1b has also been implicated in promoting tumour growth by inducing neo-angiogenesis and regulating the soluble mediators that affect tumour cell survival and spreading of metastasis [80]. IL-6 has been found to stimulate a macrophage phenotype change that promotes cancer by regulating the level of receptors for IL-4 [82]. Moreover, IL-6 can lead to increased proliferation, invasiveness, and tumour progression in the pancreatic tumour microenvironment [83]. IL-6 plays a key role in PDAC development and progression, as it affects immune suppression in the TME and enhances angiogenesis, proliferation and migration of tumour cells [84]. Elevated levels of IL-6 and IL-10 in the serum of pancreatic cancer patients have been associated with a poor prognosis [85]. Monocytes can induce NK cells to produce IL-10 [86]. IL-10 influences three important functions of the monocytes: the release of immune mediators, antigen presentation, and phagocytosis. Simply put, it suppresses all monocyte functions responsible for these cells’ positive role in innate and specific immunity [87]. Although the monocytes were more differentiated in the Low *AHR* group, the composition of surface cytokines did not have a significant impact in the Low *AHR* group of patients. A disbalance in cytokines showed a direct negative relationship between *AHR* and measured cytokine expression in PBMCs and free cytokines in the serum. The imbalance of cytokines at the transcriptomic, proteomic and circulating levels indicates that AHR is involved in several pathways that regulate anticancer function.

The identification of limitations in the current diagnostic and treatment approaches for pancreatic ductal adenocarcinoma underscores the need for future research and the development of personalized therapeutic strategies. In this context, exploring immunotherapy, potentially in conjunction with targeting the AHR-PD1/PDL1 pathway, holds promise for enhancing treatment response. Further research is necessary to understand the underlying mechanisms behind these immune features and validate the results in larger groups of PDAC patients. Ultimately, this study may contribute to the development of effective treatments for pancreatic cancer by targeting AHR signalling pathways to modulate immune responses and improve patient outcomes.

## 5. Conclusions

The findings highlight the intricate relationship between low *AHR* expression and immune dysregulation in PDAC patients, including an altered PD1/PDL1 pathway, a potential impairment in the effector arm of the immune response (T and NK cells), and significantly reduced expression of pro-inflammatory cytokines, such as *IL-1b*, *IL-6* and *IL-10*, suggesting an immunosuppressive environment at least in the peripheral blood. This might explain the checkpoint inhibitor treatment failure in a particular subset of patients and provide new opportunities for targeted immunotherapy for PDAC patients. Furthermore, targeting the AHR-PD1/PDL1 pathway, which has been implicated in immune dysregulation and the progression of PDAC, holds the potential to further enhance the effectiveness of immunotherapy.

## 6. Study Limitations

The sample size of 60 participants in this study may limit the statistical power to draw robust conclusions, and further investigations with larger cohorts are warranted to validate these findings. Additionally, it is important to consider potential confounding factors such as age, gender, and the stage of pancreatic cancer, as these variables can influence immune function and cancer progression. Lifestyle factors, including smoking, medications, and obesity, which were not accounted for in this study, may also contribute to variations in the results. Future research should address these limitations to provide a more comprehensive understanding of the relationship between *AHR* expression, immune dysregulation, and pancreatic cancer.

## 7. Practical Recommendations

In this study, we observed that a subset of PDAC patients with low expression of *AHR* showed a decreasing trend in PD1, both at the transcriptomic and proteomic levels. This finding suggests that these patients may not benefit from checkpoint inhibitor treatment targeting the PD1/PDL1 pathway. Alternative personalised approaches to immunotherapy and/or other anti-cancer treatments should be explored for this subgroup of patients. On the other hand, the elevated expression of *AHR* in PDAC patients may indicate a favourable response to checkpoint inhibitors, highlighting the potential for targeting the PD1/PDL1 pathway as a therapeutic strategy.

These findings underscore the importance of considering *AHR* expression as a potential biomarker. Ultimately, personalised treatment strategies based on *AHR* expression status could improve the outcomes of immunotherapy in PDAC patients.

## Figures and Tables

**Figure 1 cancers-15-04639-f001:**
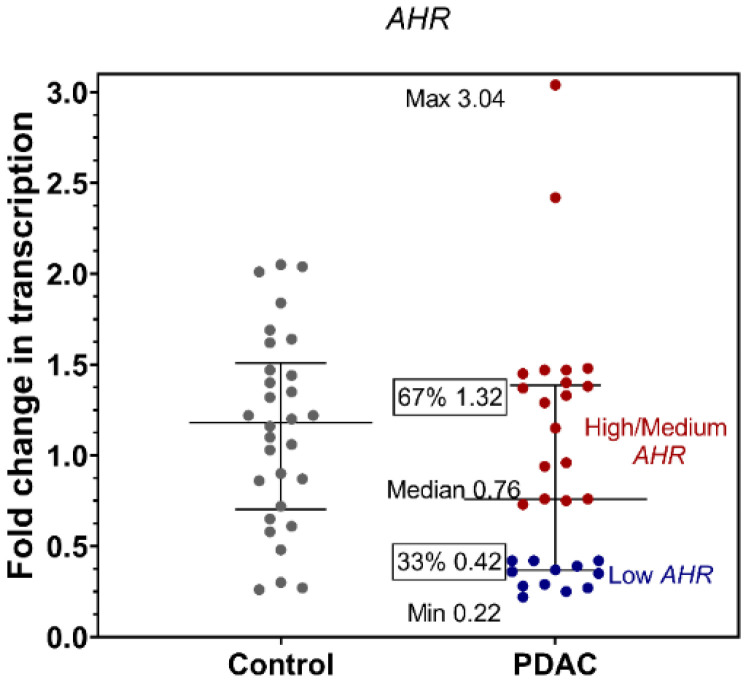
mRNA expression fold changes in the AHR gene from healthy control and PDAC patients PBMCs; PDAC patients grouping to Low *AHR* and High/Medium *AHR* by *AHR* gene expression (*p* > 0.05; Mann–Whitney test).

**Figure 2 cancers-15-04639-f002:**
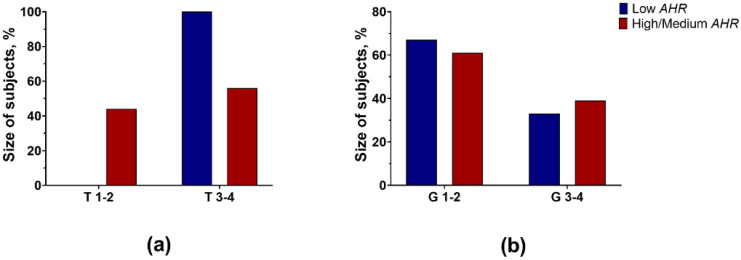
Distribution of (**a**) T stage of tumours (*p* = 0.01; Fisher’s exact test) and (**b**) differentiation grade (*p* = 1.0; Fisher’s exact test) in different groups of patients.

**Figure 3 cancers-15-04639-f003:**
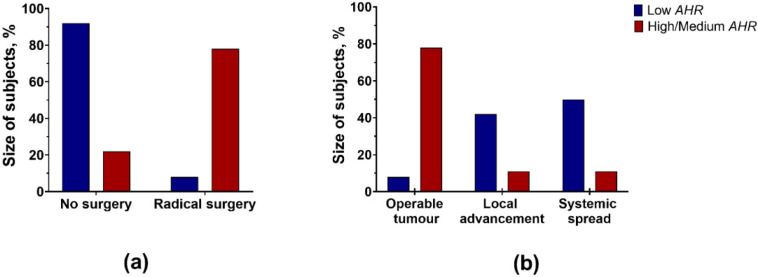
(**a**) Radical surgical treatment was contraindicated in the vast majority of Low AHR group patients (*p* < 0.001; Fisher’s exact test), (**b**) due to either systemic spread or local advancement of tumours in the Low *AHR* group of patients (*p* < 0.001; Pearson chi-square).

**Figure 4 cancers-15-04639-f004:**
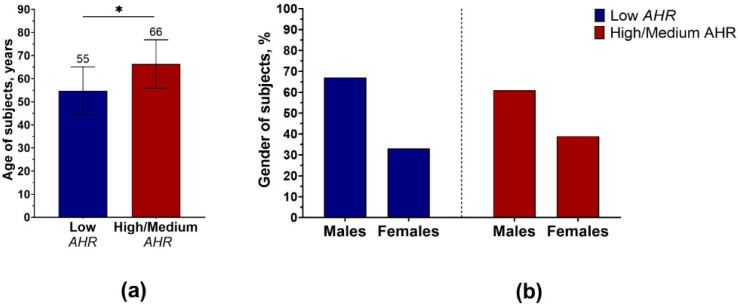
(**a**) Distribution of mean age (± SD) between pancreatic cancer patients by *AHR* level (* *p* < 0.05; Mann–Whitney). (**b**) Changes in gender percentage between pancreatic cancer patients by *AHR* level (*p* = 1.0; Fisher’s exact test).

**Figure 5 cancers-15-04639-f005:**
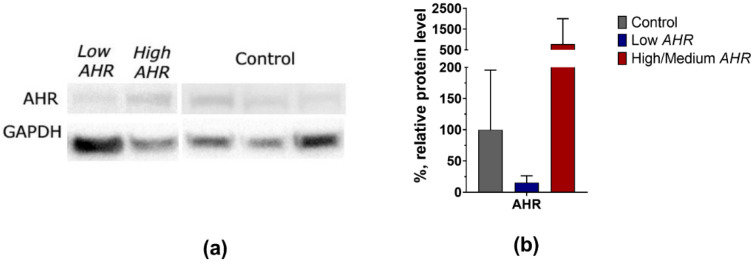
(**a**) Representative Western blots analysis of AHR from individual subjects PBMCs (1 Low *AHR*, 1 High/Medium *AHR*, 3 Ctrl). (**b**) Normalised protein level of AHR from all measured patients (4 Low *AHR*, 3 High/Medium *AHR*, 8 Ctrl) as means + SD. The uncropped bolts are shown in Appendix A.

**Figure 6 cancers-15-04639-f006:**
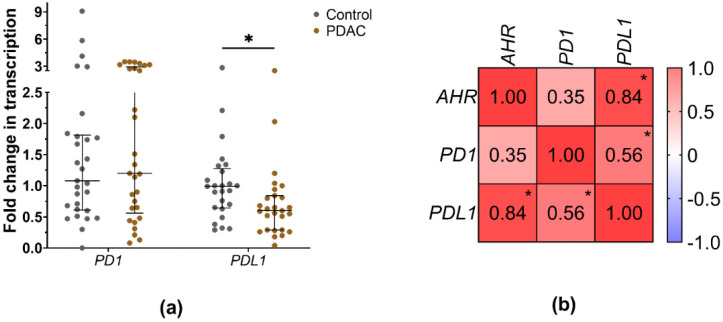
(**a**) mRNA expression of *PD1* and *PDL1* genes from healthy control and PDAC patients’ PBMCs (Mann–Whitney test, * *p* < 0.05). (**b**) *AHR*, *PD1* and *PDL1* correlation matrix between PDAC patients (Spearman correlations coefficients–r, * *p* < 0.05).

**Figure 7 cancers-15-04639-f007:**
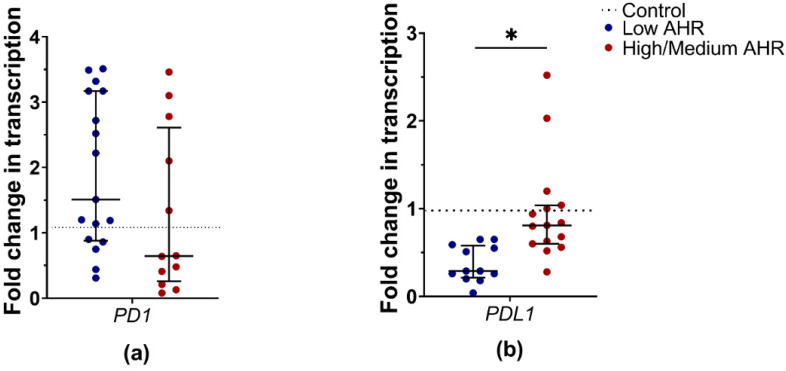
mRNA expression fold changes in (**a**) *PD1* and (**b**) *PDL1* genes from PDAC patients grouped by *AHR* expression level (Mann–Whitney test, * *p* < 0.05).

**Figure 8 cancers-15-04639-f008:**
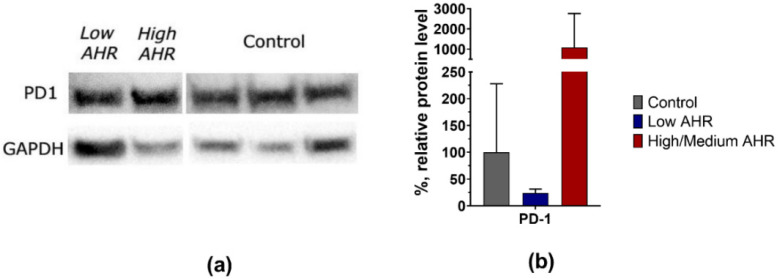
(**a**) Representative Western blots analysis of PD1 from individual subjects PBMCs (1 Low *AHR*, 1 High/Medium/Medium *AHR*, 3 Ctrl). (**b**) Normalised protein level of PD1 from all measured patients (4 Low *AHR*, 3 High/Medium *AHR*, 8 Ctrl) as means + SD. The uncropped bolts are shown in Appendix A.

**Figure 9 cancers-15-04639-f009:**
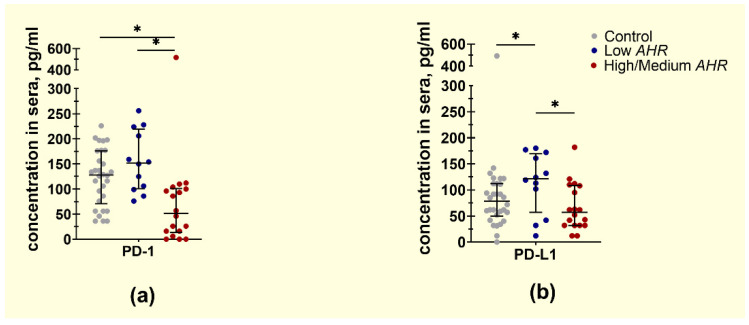
Concentration of free (**a**) PD1 and (**b**) PDL1 proteins from individual subjects’ sera by *AHR* expression level (Mann–Whitney test, pg/mL, * *p* < 0.05).

**Figure 10 cancers-15-04639-f010:**
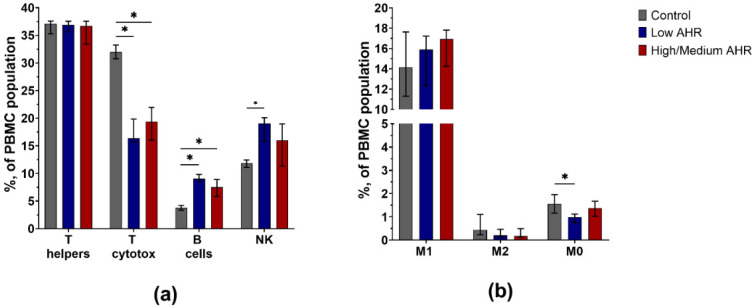
Classification of total PBMCs from healthy control and PDAC patients by *AHR* level: (**a**) lymphocyte subtypes; (**b**) monocyte subtypes (Mann–Whitney test, * *p* < 0.05 vs. Control).

**Figure 11 cancers-15-04639-f011:**
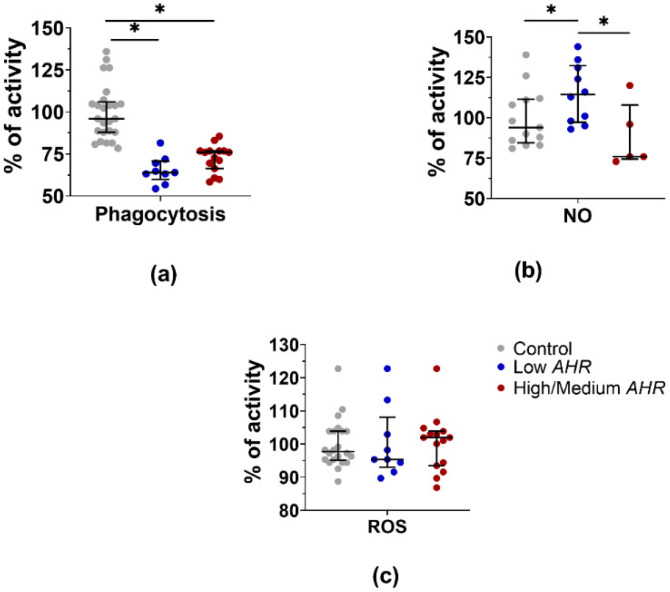
Initiated (**a**) phagocytosis, (**b**) NO production and (**c**) ROS production activity (Mann–Whitney test, * *p* < 0.05).

**Figure 12 cancers-15-04639-f012:**
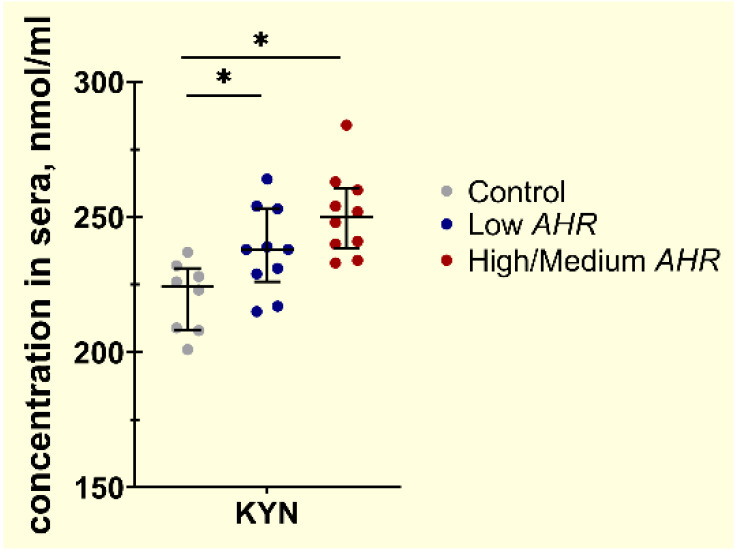
Concentration of free KYN protein from individual subjects’ sera by AHR expression level (Mann–Whitney test, nmol/mL, * *p* < 0.05).

**Figure 13 cancers-15-04639-f013:**
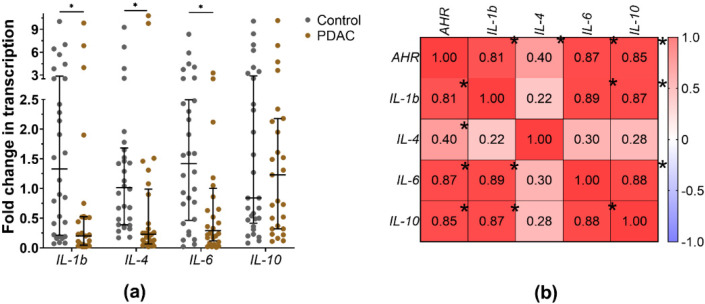
(**a**) mRNA expression fold changes in measured cytokines from PDAC patients and healthy control PBMCs (Mann–Whitney test, * *p* < 0.05), and (**b**) with *AHR* and measured cytokines correlation matrix between PDAC patients (Spearman correlations coefficients–r, * *p* < 0.05).

**Figure 14 cancers-15-04639-f014:**
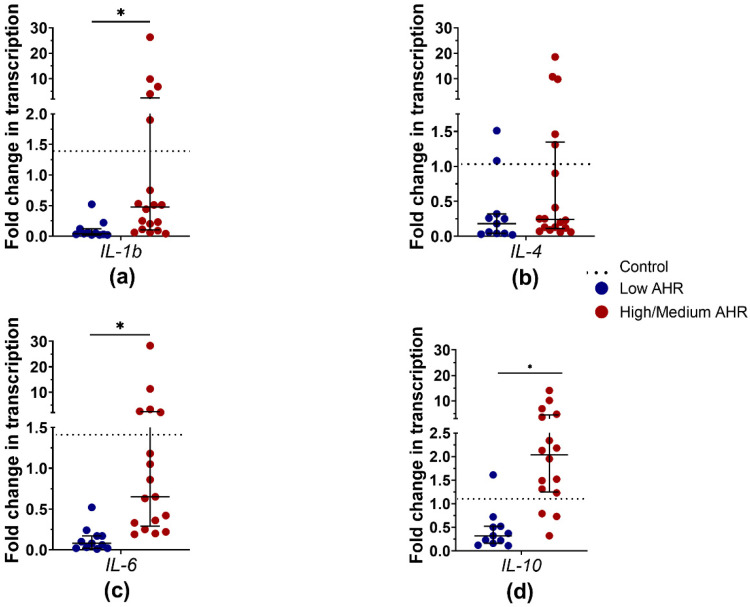
mRNA expression of measured cytokines: (**a**) *IL-1b*, (**b**) *IL-4*, (**c**) *IL-6*, (**d**) *IL-10* from PDAC patients PBMCs by *AHR* level (Mann–Whitney test, * *p* < 0.05).

**Figure 15 cancers-15-04639-f015:**
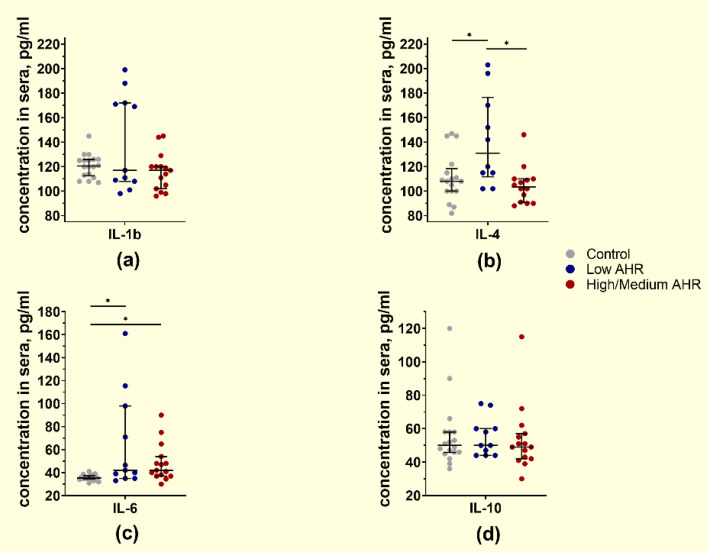
Concentration of free cytokines: (**a**) IL-1b, (**b**) IL-4, (**c**) IL-6, (**d**) IL-10 from individual subjects’ sera by *AHR* expression level (Mann–Whitney test, pg/mL, * *p* < 0.05).

**Figure 16 cancers-15-04639-f016:**
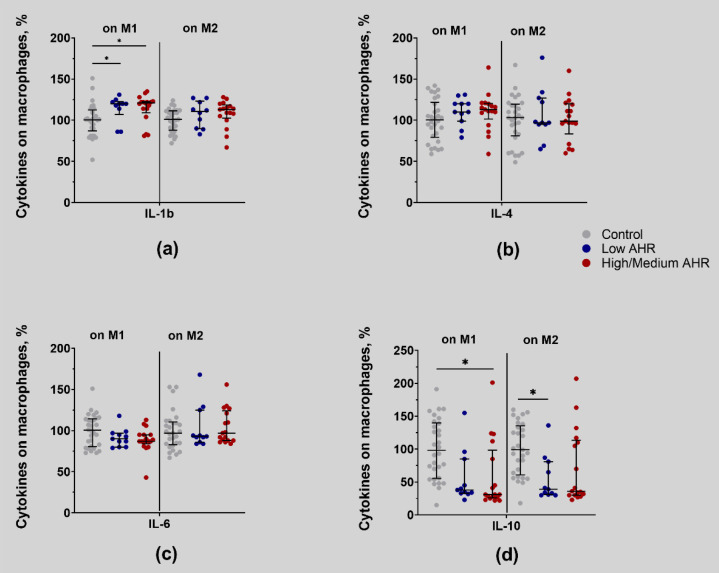
Percentage of cytokines: (**a**) IL-1b, (**b**) IL-4, (**c**) IL-6, (**d**) IL-10 on M1 and M2 type cells by *AHR* expression level (Kruskal–Wallis test, Dunn’s multiple comparisons test, * *p* < 0.05).

## Data Availability

The data presented in this study are available on request from the corresponding author.

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
