# Peer review of "Association between AHR Expression and Immune Dysregulation in Pancreatic Ductal Adenocarcinoma: Insights from Comprehensive Immune Profiling of Peripheral Blood Mononuclear Cells"

_cancers, 2023, doi:10.3390/cancers15184639_

Round 1
Reviewer 1 Report
Abstract:
The abstract is generally clear and concise. It effectively introduces the research focus, methods, and key findings.
1. The objective of the study is well-defined in the abstract. However, it is suggested to consider specifying the significance of understanding the role of AHR in pancreatic cancer immunity, as this would help readers grasp the broader implications.
2. Mentioning the number of patients in each group (Low and High/Medium AHR) is helpful. Adding a brief statement on how these groups were defined or selected could provide context.
3. The abstract provides a good summary of the key findings, such as the distinct immune features in the Low AHR group and the potential implications for anticancer immunity in PDAC. Consider mentioning the specific cytokines (other than IL-4) that showed imbalance to give readers a more detailed understanding.
4. The mention of "Further investigations are warranted" is appropriate, but it could be expanded slightly to suggest what specific questions or areas of research should be explored in these future investigations.
5. For readers unfamiliar with AHR, consider including a brief sentence explaining what the aryl hydrocarbon receptor is and why it is relevant in the context of pancreatic cancer.
Introduction:
The introduction effectively outlines the significance of PDAC and the challenges in its treatment, paving the way for discussing the potential role of immunology. Adding a bit more detail to certain points could enhance the clarity and engagement of this section.
1. Mentioning the variation in incidence and mortality rates across different regions of the world is informative. However, it might be beneficial to briefly explain why such regional differences exist, as this could spark curiosity and provide context for readers.
2. The mention of immunotherapy is relevant, but it could be more specific. You should briefly explain what immunotherapy is and why it holds promise in treating cancer in general, as well as what specific challenges have hindered its success in PDAC.
3. The introduction mentions immunological barriers, such as insufficient immune activation and excess immune suppression, as potential reasons for the limited success of immunotherapy in PDAC. While this is crucial information, it could benefit from a bit more elaboration or examples to help readers understand these barriers better.
Methods:
1. The inclusion of a control group is crucial for comparative analysis. It's good that you've mentioned that these control participants had no previous cancer history. However, the I/E criteria should be detailed.
2. I suggest to remove Table 1 and place it as a supplementary file. This would streamline the main text and make it less cluttered. However, ensure that the supplementary table is easily accessible to readers for reference.
3. Merging the PBMC isolation and cultivation sections into one can streamline the methods section and improve readability.
4. Ensure that you include proper citations or references for any manufacturer's protocols or previously established methods used in PBMC isolation and cultivation.
5. Was EDTA a suitable anticoagulant for this study, as certain anticoagulants can affect subsequent PBMC isolation and experimental outcomes?
6. Clarify which specific effectors were used to activate PBMCs and specify whether this activation targeted monocytes, lymphocytes, or both.
7. Inquire why a specific cell type purification method like MACS (Magnetic-Activated Cell Sorting) was not employed. This could help address whether working with purified monocytes or lymphocytes might have yielded different results or insights.
Results:
While the results are presented in a clear tabular format, providing additional context, discussing the significance, and ensuring consistency in reporting will enhance the reader's understanding of the findings. (DOUBLE CHECK)
Discussion and Conclusions:
Discussion and Conclusions provided are well-structured and summarizes the key findings of the study effectively. Here are some suggestions for improvement:
While the conclusion mentions the potential implications for targeted immunotherapy, it could be beneficial to elaborate a bit more on how these findings might guide future research or clinical strategies. Consider revisiting the introduction briefly to restate the significance of these findings in the context of the challenges presented earlier in the paper (e.g., late diagnosis and unsuccessful treatment strategies).
Author Response
Dear Reviewer,
Thank you for taking the time to review our manuscript. We sincerely appreciate your thorough evaluation and thoughtful comments. We have carefully considered your feedback and have made several revisions to address the issues raised. In response letter, we provide an author's note to respond to your specific points. Please see attachment.
We believe that these revisions and clarifications have significantly enhanced the quality and comprehensibility of our manuscript. We sincerely appreciate your valuable feedback and guidance throughout the review process. Please let us know if there are any further concerns or suggestions for improvement. We look forward to hearing from you.
Thank you once again for your time and consideration.
Sincerely,
Arenida Bartkeviciene

Reviewer 2 Report
The manuscript titled "Association Between AHR Expression and Immune Dysregulation in Pancreatic Ductal Adenocarcinoma: Insights from Comprehensive Immune Profiling of Peripheral Blood Mononuclear Cells" by Bartkeviciene et. al. studies the expression of AhR in pancreatic cancer patients' cultured PBMCs and associates AhR expression profiles with immune dysregulation in these patients. The cohort of patients studied is pretty small, only 30 which is a limitation of the study. However, the findings are interesting and sheds new light on the levels of AhR in pancreatic cancer patients and their poor prognosis. My concerns are below:
1. Does the low expression of AhR in patients associate with low/high levels of IDO1? IDO1 generates kynurenine which binds to AhR and causes immune suppression. The authors must check the IDO1 protein expressions in the control vs PDAC patients and try to associate them to the AhR levels.
2. What is the level of expression and activity of iNOS in PDAC patients? It has been recently reported that high nitric oxide (NO) inhibits IDO1 while low NO greatly activates IDO1 (PMID: 37116709). Does high IDO1 activity contribute to the immune dysregulation in PDAC patients? This must be checked and discussed in the Discussion section.
I recommend the manuscript for a major revision. Thank you.
Author Response

(The authors gave the same response as above.)

Round 2
Reviewer 2 Report
The authors have considered my suggestions and have reasonably improved the manuscript. I would recommend accepting the revised manuscript for publication. Thank you.